# Promiscuous Lipase-Catalyzed Markovnikov Addition of H-Phosphites to Vinyl Esters for the Synthesis of Cytotoxic α-Acyloxy Phosphonate Derivatives

**DOI:** 10.3390/ma15051975

**Published:** 2022-03-07

**Authors:** Paweł Kowalczyk, Dominik Koszelewski, Barbara Gawdzik, Jan Samsonowicz-Górski, Karol Kramkowski, Aleksandra Wypych, Rafał Lizut, Ryszard Ostaszewski

**Affiliations:** 1Department of Animal Nutrition, The Kielanowski Institute of Animal Physiology and Nutrition, Polish Academy of Sciences, Instytucka 3, 05-110 Jabłonna, Poland; 2Institute of Organic Chemistry PAS, Kasprzaka 44/52, 01-224 Warsaw, Poland; jan.samsonowicz-gorski.stud@pw.edu.pl (J.S.-G.); ryszard.ostaszewski@icho.edu.pl (R.O.); 3Institute of Chemistry, Jan Kochanowski University, Uniwersytecka 7, 25-406 Kielce, Poland; b.gawdzik@ujk.edu.pl; 4Department of Physical Chemistry, Medical University of Bialystok, Kilińskiego 1 Str., 15-089 Białystok, Poland; kkramk@wp.pl; 5Centre for Modern Interdisciplinary Technologies, Nicolaus Copernicus University, Torun ul. Wileńska 4, 87-100 Toruń, Poland; wypych@umk.pl; 6Institute of Mathematics, Informatics and Landscape Architecture, The John Paul II Catholic University of Lublin, ul. Konstantynów 1 H, 20-708 Lublin, Poland; lizut@kul.pl

**Keywords:** α-acyloxy phosphonates, enzymatic Markovnikov addition, lipase promiscuity, Fpg protein-formamidopyrimidine, lipopolysaccharide (LPS), HIV-human immunodeficiency virus

## Abstract

An enzymatic route for phosphorous-carbon- bond formation is developed by discovering new promiscuous activity of lipase. This biocatalytic transformation of phosphorous-carbon- bond addition leads to biologically and pharmacologically relevant α-acyloxy phosphonates with methyl group in α-position. A series of target compounds were synthesized with yields ranging from 54% to 83% by enzymatic reaction with *Candida cylindracea* (CcL) lipase via Markovnikov addition of *H*-phosphites to vinyl esters. We carefully analyzed the best conditions for the given reaction such as the type of enzyme, temperature, and type of solvent. The developed protocol is applicable to a range of *H*-phosphites and vinyl esters significantly simplifying the preparation of synthetically challenging α-pivaloyloxy phosphonates. Further, the obtained compounds were validated as new potential antimicrobial drugs with characteristic *E. coli* bacterial strains and DNA modification recognized by the Fpg protein, *N*-methyl purine glycosylases as new substrates. The impact of the methyl group located in the α-position of the studied α-acyloxy phosphonates on the antimicrobial activity was demonstrated. The pivotal role of this group on inhibitory activity against selected pathogenic *E. coli* strains was revealed. The observed results are especially important in the case of the increasing resistance of bacteria to various drugs and antibiotics.

## 1. Introduction

α-Hydroxy phosphonates are recognized as a valuable class of substances exhibiting interesting medicinal and agricultural properties [1,2,3,4]. α-Hydroxy phosphonate derivatives are of great importance due to their various biological actions and therapeutic applications such as bactericidal [5], fungicidal [6], antitumor [7], and HIV inhibitory activities [8]. The high interest in this class of compounds has been reflected in numerous publications on their synthesis, modification, and application not only in medical chemistry [9,10,11,12,13,14]. It was shown that further functionalization of the hydroxy group has a pivotal impact on the biological activity of the modified α-hydroxy phosphonates. For example, 2-dimethylphosphonoethyl acetate which was synthesized via radical addition reaction of *H*-phosphite to vinyl acetate exhibits broad bioactivity, e.g., anti-hypoxic, soporific, anticonvulsive, and hypothermic (Figure 1) [15].

Due to the large number of antimicrobial resistances, there is a growing need to elaborate lead structural scaffolds that may be useful in developing potent antimicrobial drug. Furthermore, it was noticed that the antitumor activity of many chemotherapeutic compounds is directly correlated with their antibiotic features, and α-hydroxy and α-acyloxy phosphonate derivatives are still being rediscovered in various angles in practical clinical biochemistry and microbiological engineering. Revising toxic effect on bacterial cells of the novel organic compounds containing a phosphorus–carbon bond can provide suitable antimicrobial agents against bacterial clinical pathogens. A careful analysis of the literature data showed that there are no data related to the influence of the size of the substituent particularly methyl group in the alpha position of the α-hydroxy phosphonate derivatives on their antimicrobial activity. Unfortunately, there are no mild and environmentally sustainable methods to synthesize these types of compounds. The available methods have disadvantages that often limit their use in the synthesis of biologically active compounds. An important aspect that must be met in the synthesis of such compounds is the lack of metal impurities, the content of which, in accordance with pharmacological requirements, cannot exceed 5 ppm. The goal of the presented studies is the development of efficient enzymatic preparation of α-acetoxy phosphonate derivatives with methyl in the α-position, and their validation and comparison with α-aryl substituted derivatives as antimicrobial agents against model strains of *Escherichia coli* K12 (with native LPS in its structure) and R2–R4 (LPS of different lengths in its structure).

## 2. Materials and Methods

### 2.1. Microorganisms and Media

All bacterial strains were from Prof. Jolanta Łukasiewicz (Polish Academy of Sciences, Wrocław, Poland). Reagents and apparatus that were used in the work were described in detail in our earlier works in this field.

The synthesis of new compounds from the α-hydroxy phosphonate derivatives, similarly to other compounds studied by us in previous works, may determine a new alternative to the commonly used antibiotics in clinical infections. This chemical and biological activity is related to two kinds of specific substituents in their structure R1 and R2 (see Figure 1). Dysfunction of bacterial membranes containing different lengths of LPS in model bacterial strains is an ideal model to assess the effectiveness of these compounds in relation to the antibiotics used by a specific enzyme Fpg of modified bacterial DNA (Labjot, New England Biolabs, Ipswich, MA, USA).

### 2.2. Statistical Analysis

All experimental data were presented as the means ± standard error of the mean (SEM, manufacturer, Saint Louis, MO, USA) of a minimum of three independent experiments (n = 3). The Tukey test was used Statistical significance is indicated by (*p* < 0.05): * *p* < 0.05, ** *p* < 0.1, *** *p* < 0.01.

### 2.3. General Methods of Synthesis α-Hydroxy Phosphonate Derivatives

All reagents and the solvents were from Sigma-Aldrich (Saint Louis, MI, USA). NMR spectra (Saint Louis, MI, USA) were reordered on Varian apparatus (Varian, Saint Louis, MI, USA) (400 MHz), mass spectrometer was from Waters Company, Milford, USA. All specific strains such as *Pseudomonas fluorescens* (PfL) (catalogue number 534730, Lot. number MKBH1198V), *Candida rugosa* (CrL) (catalogue number 90860, Lot. number BCBH7102V), *Candida cylindracea* (CcL) (catalogue number 62316, Lot. number 1336707), and Bovine serum albumin were purchased from Sigma-Aldrich. Immobilized lipase from *Candida antarctica* B (Novozym 435) (catalogue number LC200223) was purchased from Novo Nordisk (Warsaw, Poland). Lipases from porcine pancreas, Type II (PpL) (catalogue number L-3126, Lot. number 108H1379) was purchased from Sigma-Aldrich.Dimethyl 1-hydroxy-1-phenylmethylphosphonate (**1**) and dimethyl hydroxy-(2-nitrophenyl)methylphosphonate (**3**) were obtained previously [16]. 

*Dimethyl Hydroxy-(2-nitrophenyl)methylphosphonate* (**2**): Compound **2** was obtained according to literature procedure [16] with 69% yield (180 mg, 0.69 mmol) as a colorless solid; m.p. 166–167 °C [Lit 166.5 °C] [17]; ^1^H NMR (400 MHz, CDCl_3_) δ 8.03–7.95 (m, 2H), 7.71–7.63 (m, 1H), 7.49–7.41 (m, 1H), 6.27 (dd, *J* = 14.0, 5.9 Hz, 1H), 5.59 (t, *J* = 7.0 Hz, 1H), 3.74 (d, *J* = 2.2 Hz, 3H), 3.71 (d, *J* = 2.5 Hz, 3H); ^31^P NMR (162 MHz, CDCl_3_) δ 22.22. NMR data were in accordance with those reported in the literature [18]; LRMS (ESI) m/z calcd for C_9_H_13_NO_6_P [M + H]^+^ 262.1, found 262.1. 

*Dibenzyl (1-Hydroxy-2-phenylpropyl)phosphonate* (**4**). Compound **4** was obtained as a mixture of diastereoisomers at 2:1 ratio according to literature procedure [16] with 53% yield (210 mg, 0.53 mmol) as a colorless semi-solid; ^1^H NMR (400 MHz, CDCl_3_) δ 7.43–7.15 (m, 23H), 5.05–4.86 (m, 5H), 4.80 (dd, *J* = 11.8, 8.9 Hz, 1H), 4.10 (td, *J* = 6.2, 1.0 Hz, 1H), (diastereoisomers) 3.33–3.19 (m, 2H), 1.45 (d, J = 7.1 Hz, 3H), (diastereoisomer) 1.40 (d, *J* = 7.2 Hz, 1H); ^31^P NMR (162 MHz, CDCl_3_) δ 24.99, 24.72. HRMS (EI) m/z calcd for C_23_H_26_O_4_P [M + H]^+^ 397.1569, found 397.1549.

#### General Procedure for the Synthesis of α-Hydroxy Phosphonate Derivatives **5**–**9** and Phosphonate **10**

General procedure for enzyme catalyzed addition reaction of *H*-phosphites to vinyl esters. A mixture of the corresponding vinyl ester or dimethyl maleate (1 mmol), *Candida cylindracea* lipase (CcL, 80 mg) and the corresponding *H*-phosphite (1 mmol) in *n*-hexane (2 mL) was shaken at 200 rpm at 40 °C for 18 h. After the completion of reaction. the catalyst was separated on a glass frit funnel. The residue was washed with ethyl acetate. The combined organic phase was concentrated under vacuum. The resulting residue was purified by column chromatography (silica gel, eluent: ethyl acetate/hexanes, 6:4) to afford the target α-hydroxy phosphonate derivatives **5**–**9** and phosphonate **10**. The yields of the derivatives are shown in Figure 2. The structures of all compounds were confirmed using NMR and mass spectroscopy. 

*1-(Dimethoxyphosphoryl)ethyl cinnamate* (**5**). Compound **5** was obtained according to the general procedure as colorless oil with 61% yield (173 mg, 0.61 mmol); ^1^H NMR (400 MHz, CDCl_3_) δ 7.74 (d, *J* = 16.0 Hz, 1H), 7.57–7.48 (m, 2H), 7.44–7.32 (m, 3H), 6.47 (d, *J* = 16.0 Hz, 1H), 5.45 (dq, *J* = 8.5, 7.1 Hz, 1H), 3.82 (dd, *J* = 10.6, 6.7 Hz, 6H), 1.55 (dd, *J* = 16.8, 7.1 Hz, 3H); ^31^P NMR (162 MHz, CDCl_3_) δ 24.11. NMR data were in accordance with those reported in the literature [19]. LRMS (ESI) m/z calcd for C_13_H_17_O_5_P [M + H]^+^ 285.1, found 285.1.

*1-Benzoyloxy-1-diethylphosphonylethane* (**6**). Compound **6** was obtained according to the general procedure as colorless oil with 77% yield (210 mg, 0.77 mmol); ^1^H NMR (400 MHz, CDCl_3_) δ 8.13–7.99 (m, 2H), 7.63–7.52 (m, 1H), 7.52–7.38 (m, 2H), 5.52 (dq, *J* = 8.6, 7.1 Hz, 1H), 4.28–4.11 (m, 4H), 1.58 (dd, *J* = 16.7, 7.1 Hz, 3H), 1.32 (td, *J* = 7.1, 1.5 Hz, 6H); ^13^C NMR (100 MHz, CDCl_3_) δ 165.3, 133.3, 129.8, 129.5, 128.4, 65.8, 64.1, 62.9, 62.8, 62.7, 62.6, 16.5, 16.4, 16.3, 15.2; ^31^P NMR (162 MHz, CDCl_3_) δ 21.38. HRMS (EI) m/z calcd for C_13_H_19_O_5_P [M + H]^+^ 286.0970, found 286.0977.

*Dibenzyl(1-propionoxy ethyl)phosphonat* (**7**). Compound **7** was obtained according to the general procedure as colorless oil with 83% yield (301 mg, 0.83 mmol); ^1^H NMR (400 MHz, CDCl_3_) δ 7.38–7.16 (m, 10H), 5.31 (dq, *J* = 8.5, 7.1 Hz, 1H), 5.17–4.92 (m, 4H), 2.36–2.18 (m, 2H), 1.44 (dd, *J* = 17.1, 7.1 Hz, 3H), 1.07 (t, *J* = 7.5 Hz, 3H); ^13^C NMR (100 MHz, CDCl_3_) δ 173.1, 135.9, 128.6, 128.5, 128.5, 128.4, 127.9, 127.8, 68.1, 68.0, 68.0, 67.9, 65.2, 63.5, 27.3, 15.0, 8.9; ^31^P NMR (162 MHz, CDCl_3_) δ 22.68. HRMS (EI) m/z calcd for C_19_H_24_O_5_P [M + H]^+^ 363.1361, found 363.1361.

*2,2-Dimethylpropanyloxy1-dibenzylphosphonylethane* (**8**). Compound **8** was obtained according to the general procedure as colorless oil with 54% yield (211 mg, 0.54 mmol); ^1^H NMR (400 MHz, CDCl_3_) δ 7.36–7.30 (m, 10H), 5.32 (dq, *J* = 8.2, 7.1 Hz, 1H), 5.05 (ddd, *J* = 12.1, 8.0, 1.4 Hz, 4H), 1.43 (dd, *J* = 17.1, 7.0 Hz, 3H), 1.15 (s, 9H); ^13^C NMR (100 MHz, CDCl_3_) δ 177.0, 136.0, 128.5, 127.9, 127.7, 126.9, 68.0, 67.9, 63.3, 38.7, 26.9, 14.9; ^31^P NMR (162 MHz, CDCl_3_) δ 22.67. HRMS (EI) m/z calcd for C_21_H_28_O_5_P [M + H]^+^ 391.1674, found 391.1669. 

*1-(Dibenzylphosphoryl)ethyl dodecanoate* (**9**). Compound **9** was obtained according to the general procedure as colorless oil with 81% yield (396 mg, 0.81 mmol); ^1^H NMR (400 MHz, CDCl_3_) δ 7.41–7.27 (m, 10H), 5.32 (dq, *J* = 8.7, 7.1 Hz, 1H), 5.16–4.98 (m, 4H), 2.33–2.16 (m, 2H), 1.55 (t, *J* = 7.2 Hz, 2H), 1.44 (dd, *J* = 17.1, 7.1 Hz, 3H), 1.35–1.20 (m, 17H), 0.88 (t, *J* = 6.8 Hz, 3H; ^13^C NMR (100 MHz, CDCl_3_) δ 172.4, 136.1, 128.5, 128.5, 128.4, 128.4, 127.9, 127.9, 127.9, 127.8, 68.0, 68.0, 67.9, 65.1, 63.4, 34.0, 31.8, 30.8, 29.5, 29.4, 29.3, 29.2, 29.1, 29.0, 24.8, 22.6, 15.0, 14.0; ^31^P NMR (162 MHz, CDCl_3_) δ 22.72. HRMS (EI) m/z calcd for C_28_H_42_O_5_P [M + H]^+^ 489.2770, found 489.2764.

*Dimethyl 2-(dibenzyloxyphosphoryl)succinate* (**10**). Compound **10** was obtained according to the general procedure as colorless oil with 68% yield (276 mg, 0.68 mmol); ^1^H NMR (400 MHz, CDCl_3_) δ 7.45–7.29 (m, 10H), 5.14–4.95 (m, 4H), 3.70 (d, *J* = 0.7 Hz, 3H), 3.64 (s, 3H), 3.53 (ddd, *J* = 24.3, 11.1, 3.7 Hz, 1H), 3.06 (ddd, *J* = 17.5, 11.1, 8.1 Hz, 1H), 2.79 (ddd, *J* = 17.5, 9.8, 3.7 Hz, 1H); ^13^C NMR (100 MHz, CDCl_3_) δ 171.2, 168.4, 135.7, 128.6, 128.6, 128.5, 127.9, 68.4, 68.4, 68.3, 52.8, 52.1, 42.0, 40.7, 31.1, 31.1; ^31^P NMR (162 MHz, CDCl_3_) δ 22.05. HRMS (EI) m/z calcd for C_20_H_24_O_7_P [M + H]^+^ 407.1260, found 407.1263.

## 3. Results

### 3.1. Chemistry

Organophosphorus compounds display a broad spectrum of various important biological activities [20]. Among others, α-hydroxy phosphonates have grasped attention towing to their wide range bioactivity. α-hydroxy phosphonates are usually synthesized by the base-catalyzed Pudovik reaction, the addition of dialkyl phosphite to an oxo compound [4,21]. Recently, we have shown that this reaction can be catalyzed by lipases providing desired products with high yield under environmentally sustainable conditions [16]. α-acyloxy phosphonates, an acyl derivative of α-hydroxy phosphonates known to exhibit potential bioactivity as herbicides [19,22,23,24], anticancer agents [25], fungicides [26], and insecticides [20]. Several articles described the acylation of α-hydroxy phosphonates with carboxylic acid chlorides using either triethylamine or pyridine [27,28]. Moreover, acetic anhydride has been used with copper(II) trifluoromethanesulfonate or trichlorotitanium(IV) trifluoromethanesulfonate as a catalyst [29,30]. Carboxylic acids can be also used as an acylating agents for α-hydroxy phosphonates supported by coupling reagents e.g., *N*,*N*′-dicyclohexylcarbodiimide and 4-dimethylaminopyridineas as the base [31]. Apart from Michaelis–Arbuzov and Michaelis–Becker reactions, the phosphorus–carbon bond is formed using phospha-Michael addition under basic conditions [32,33,34,35,36,37]. Therefore, the possibility of using this approach in the synthesis of bioactive α-acyloxy phosphonates should be considered. Although some chemical strategies based on addition of *H*-phosphites towards the synthesis of α-acyloxy phosphonates catalyzed either by sodium ethanolate or K_2_CO_3_/18-crown-6 ether have been reported [38,39], the biocatalytic synthesis of target α-acyloxy phosphonates via addition of *H*-phosphites to unsaturated carbon–carbon bond remains unexploited.

The use of hydrolases as catalysts in various variants of Michael addition of different nucleophilic partners was widely discussed in the literature [40,41,42,43,44]. Recently, it was also shown that lipases catalyze selectively Markovnikov additions of thiols and amines to vinyl esters in different organic solvents [45,46,47,48]. Consequently, recognition of the new promiscuous activities of hydrolases provides novel processes to access pharmaceutically effective compounds [49]. 

As a continuation of our research on sustainable protocols, [50,51,52,53] we paid attention to find an environmentally friendly method for desired α-hydroxy phosphonates **1**–**4** [16] and α-acyloxy phosphonates **5**–**10** (Figure 2). 

Based on literature data regarding enzymes known—bioactivity [54], the model addition reaction of dimethyl phosphite (1 mmol) and vinyl cinnamate (1 mmol) was conducted in *n*-hexane at 40 °C (Figure 1) (Table 1, entry 1). A series of commercially available lipases and one domestically prepared pig liver acetone powder were screened as catalysts. The results are summarized in Table 1.

From above enzymatic screening, CcL was found as the best catalyst among the lipases checked for this addition reaction (Table 1, entry 2). The progress of the formation of target product **5** was monitored by TLC (thin layer chromatography). In less than 10 hours, the α-acyloxy phosphonate **5** was obtained in good yield (57%), and no significant increase in reaction yield was noticed after 18 hour (61%). In the blank reaction, without enzyme, no reaction proceeded (Table 1, entry 1). To confirm the promiscuous activity of CcL in Markovnikov addition, non-catalytic bovine serum albumin (BSA) [55] and thermally deactivated CcL were used. The results indicated that BSA and denatured CcL gave only traces of product **5** (Table 1, entries 14 and 16). These results clearly show that the peculiar active site of CcL is responsible for the studied addition reaction of *H*-phosphite to vinyl ester.

Considering the impact of solvent on enzyme stability it can change reactivity of enzyme [56]. To find the efficient medium, we conducted the model reaction in various solvents; *tert*-butyl methyl ether (TBME), dimethyl sulfoxide (DMSO), tetrahydrofuran (THF), dichloroethane (DCM), and toluene. The obtained results revealed that the solvent have significant effects on CcL activity. As shown in Table 1, the lipase from *Candida cylindracea* was active in all tested solvents, and the non-protic solvents such as *n*-hexane and TBME were found to be more suitable. Therefore, *n*-hexane was applied in the following process. Obtained results remain in agreement with the literature data for enzymatic aza-Michael addition [54]. In the next step, the impact of enzyme quantity was investigated. A slight increase in yield of target compound 5 was noticed by raising the amount of CcL from 80 mg to 100 mg. Thus, the further reactions were conducted with 80 mg of CcL. Enzyme activity and stability are strongly associated with the temperature of reactions [52]. To verify the effect of temperature, we conducted model reaction in temperature from 30 °C to 50 °C; however, the yield was reduced above 40 °C, which may be due to changes in the quaternary structure of the used enzyme (Table 1, entries 8 and 9). It is noteworthy to mention that the studied addition reaction catalyzed by CcL afforded no by-products resulting from anti-Markovnikov addition or hydrolysis. Unfortunately, obtained chiral product exhibited low or no optical activity what remains in agreement with literature data regarding lipase catalyzed Michael-type additions [43] or Markovnikov addition of thiols and amines to vinyl esters [45,46,47,48]. Due to the fact that the target α-acyloxy phosphonates (Figure 2) under studied conditions can be formed in two different ways: by the direct addition of *H*-phosphite to the double bond or by the enzymatic cascade reaction involving α-hydroxy phosphonate synthesis [16] from acetaldehyde derived from vinyl ester hydrolysis via enzyme catalyzed Pudovik reaction and further esterification with carboxylic acid. Additional experiments under reaction course were performed. Under the developed conditions, neither the formation of the α-hydroxy phosphonate nor enzymatic esterification was noticed. The obtained results allow to state that we are dealing with a direct *H*-phosphite addition reaction to vinyl ester.

The reusability of an enzyme is an important feature that significantly increases the attractiveness of the developed method by significantly reducing overall costs. In this work, *Candida cylindracea* lipase was reused up to four times with gradual decrease of the yield to 39% after four cycles.

Finally, we explored the developed reaction conditions to various vinyl esters and *H*-phosphites which resulted in α-acyloxy phosphonate derivatives **5**–**9** with moderate to high yields (Figure 2). Developed protocol provides straightforward and direct access to α-acyloxy phosphonates **5** and **6**, which are structural analogs of compounds with documented herbicidal and antitumor activities (Figure 1). The Markovnikov addition to sterically hindered vinyl pivalate provided product **8** with lower yield (54%, Figure 2). The addition of *H*-phosphonates to vinyl benzoate and vinyl cinnamate provided products with slightly lower yields than with aliphatic vinyl esters. It can be because of weak nucleophilic character of aromatic acid vinyl esters compared to aliphatic ones. To our delight, the developed conditions of the enzymatic addition reaction allowed for the phospha-Michael addition of *H*-phosphonate to dimethyl maleate, what resulted in phosphonate **10** with 68% yield (Figure 2). This opens up new possibilities for the use of lipase promiscuity in the formation of phosphorus-carbon bonds. The structures of compounds **2**, **4** and **5**–**10** are presented in experimental part (Appendix A).

### 3.2. Cytotoxic Studies of the Library of α-Hydroxy Phosphonate Derivatives

The obtained results show that all tested α-hydroxy phosphonate derivatives have a cytotoxic effect in all analyzed *E. coli* bacterial strains differing in LPS length. Different inhibitory activity was noted depending on the nature of the substituent in the α-position of the tested compounds. All tested α-acyloxy phosphonates 5–7 with methyl group at α-position exhibited higher activity against strains R2, R3, and R4 than commonly used antibiotics (Figure 3, Figure 4, Figure 5, Figure 6 and Figure 7). The MIC and MBC test values for each model of *E. coli* R2–R4 and K12 strains were visible in all analyzed growth microplates after the addition of resazurin.

Model strains of *E.coli* were plotted in all 48-well plates observed: K12, R2-R4 which were treated with the analyzed compounds. From analysis of the MIC and MBC assays, color changes were observed for all compounds tested but at different levels and at different dilutions. Bacterial strains R3 and R4 were the most susceptible to modification with these compounds due to the increasing length of their LPS (visible dilutions of 10^−2^ corresponding to a concentration of 0.0015 µM) than strains K12 and R2 (visible dilutions of 10^−6^ corresponding to a concentration of 0.02 µM). The analyzed R4 strain was the most sensitive of all strains, probably due to the longest length of the lipopolysaccharide chain in the bacterial membrane. In all analyzed cases, the MBC values were approximately 50 times higher than the MIC values (Figure 3, Figure 4, Figure 5, Figure 6 and Figure 7 and Table 2).

### 3.3. Analysis of Bacterial DNA Isolated from E. coli R2–R4 Strains Modified with α-Hydroxy Phosphonate Derivatives

The analyzed compounds were selected for further analyzes by modifying them with the bacterial DNA obtained from the analyzed strains. Modified bacterial DNA was digested by using Fpg protein. We wanted to observe whether the resulting modifications in bacterial DNA would introduce oxidative damage to the DNA chain by changing the topological three forms [50,57,58,59,60,61,62]. Probably the size of the specific substituent at α-position may determine the toxicity of the analyzed *E. coli* strains, including in particular R4, as evidenced by the obtained MIC and MBC values. The obtained results for individual compounds were statistically significant at the level of *p* < 0.05. (Figure 6).

### 3.4. Modification of Plasmid DNA Isolated from E. coli R2–R4 Strains with Tested α-Hydroxy Phosphonate Derivatives

Performed studies proved that the analyzed and newly synthesized compounds can potentially be used as “substitutes” for the currently used antibiotics in hospital and clinical infections (Figure 7 and Figure 8 and Appendix A).

Large modifications of plasmid DNA were observed for all analyzed compounds, especially for those marked with numbers **1**–**4** and **10** showed super selectivity in all analyzed bacterial strains, even differentiating the cytotoxicity in the K12 strain (See Appendix A).

## 4. Conclusions

A novel enzymatic route towards target α-acyloxy phosphonates which omits usage of toxic catalysts was develop. For the first time, a new and non-natural activity of lipases was shown in a Markovnikov addition reaction leading to the bioactive α-acyloxy phosphonates in organic solvent. The protocol developed was used for the synthesis of series target products possessing methyl group at alpha position. This advanced activity of lipases is of fundamental importance in expanding the application of enzymes and in the evolution of new biocatalysts. Our discovered protocol offers facile metal-free synthesis, environmental sustainability, easy work-up procedure, and high yields of the target products (54–83%). Toxic effect of the obtained phosphonate derivatives was evaluated on model *E.coli* strains. The analyzed phosphonate derivatives are able to modify all model strains of *E. coli* (R2-R4) and their bacterial DNA, changing the spatial structure of the LPS contained in their cell membranes. Among the derivatives studied, the compounds possessing methyl group at α-position **5**, **6**, **7**, **8,** and **9** turned out to be the most active comparing to derivatives with aryl group at α-position (Figure 2). The research results presented are important for understanding the biological properties of tested α-hydroxy phosphonate derivatives as a function of potential new antibiotics and their toxic effects on Gram-negative bacteria in the face of the growing drug resistance pandemic, in reference to our previous works related to the characteristics of the model *E.coli* K12 and R2-R4 [44,51,52,53,54,55,56,57,58,59,60,61,62,63,64,65]. Finally, the analyzed α-hydroxy phosphonate derivatives are more cytotoxic in the model bacterial cells than the commonly used antibiotics: ciprofloxacin, bleomycin, and cloxacillin.

## Data Availability

On request by those interested.

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
