# Peer review of "Promiscuous Lipase-Catalyzed Markovnikov Addition of H-Phosphites to Vinyl Esters for the Synthesis of Cytotoxic α-Acyloxy Phosphonate Derivatives"

_materials, 2022, doi:10.3390/ma15051975_

Round 1
Reviewer 1 Report
The manuscript deals with A new enzymatic, promiscuous procedure for lipase catalyzed phosphorous-carbon-bond addition leading to biologically and pharmacologically relevant α-acyloxyphosphonates. The authors have also validated the synthesized compounds as new potential antimicrobial drugs with characteristic E.coli bacterial strains and DNA modification recognized by the Fpg protein, N-methyl purine glycosylases as new substrates. As understood that this novel Markovnikov addition activity of lipases is of practical significance in expanding the application of enzymes and in the evolution of new biocatalysts.
It is understandable that the authors have well arranged the content of the paper in such a way that appeals to a broad range of readers. I support its publication with the following minor corrections/suggestions.
- The sentence on Page-2, Line 53 “Revising toxic effect on bacterial cells of 53 the novel organic compounds containing a phosphorus–carbon bond provides can provide an suitable antimicrobials agents against bacterial clinical pathogen” has to be revised.
- Introduction of the manuscript may be improved appropriately with more citations.
- Authors should provide H-NMR of few final products in order to provide the conformity of the products.
- Melting point and elemental analysis of the compounds should be provided wherever possible.
- The fonts on the X-Y axis of the Figures 3-4 are in light color. It shall be darker appropriately.
Author Response
Thank you very much for all the valuable comments that contributed to the improvement of the quality of the manuscript
Reviewer 1
The manuscript deals with A new enzymatic, promiscuous procedure for lipase catalyzed phosphorous-carbon-bond addition leading to biologically and pharmacologically relevant α-acyloxyphosphonates. The authors have also validated the synthesized compounds as new potential antimicrobial drugs with characteristic E.coli bacterial strains and DNA modification recognized by the Fpg protein, N-methyl purine glycosylases as new substrates. As understood that this novel Markovnikov addition activity of lipases is of practical significance in expanding the application of enzymes and in the evolution of new biocatalysts.
It is understandable that the authors have well arranged the content of the paper in such a way that appeals to a broad range of readers. I support its publication with the following minor corrections/suggestions.
- The sentence on Page-2, Line 53 “Revising toxic effect on bacterial cells of 53 the novel organic compounds containing a phosphorus–carbon bond provides can provide an suitable antimicrobials agents against bacterial clinical pathogen” has to be revised.
Response: We are grateful for this remark. Mentioned sentence was revised and corrected.
- Introduction of the manuscript may be improved appropriately with more citations.
Response: We are grateful for this remark. According to the Reviewer suggestion Introduction was revised and modified it was also supplemented with more citations
- Authors should provide H-NMR of few final products in order to provide the conformity of the products.
Response: We are grateful for this remark. NMR spectra of all studied compound together with high resolution mas spectra are provided in Supplementary Material attached to this Manuscript.
- Melting point and elemental analysis of the compounds should be provided wherever possible.
Response: Thank you for this remark. Melting point were provided for solid products. In most cases target products were obtained as colourless thick oils. All obtained compound were characterized in accordance with the prevailing requirements for this and other journals (Materials 2021, 14(19), 5725; https://doi.org/10.3390/ma14195725; Materials 2021, 14(11), 2956; https://doi.org/10.3390/ma14112956; Materials 2020, 13(22), 5169; https://doi.org/10.3390/ma13225169). Carbon, proton and phosphor NMR spectra were made, which confirm the purity and structure of the compound, additionally high resolution mass spectra were made.
- The fonts on the X-Y axis of the Figures 3-4 are in light color. It shall be darker appropriately
Response: All the drawings have been corrected into clear blue arrows on the x and y axes. We use certain bright colors as a specific code which we have adopted in other publications of our authorship [50, 57-62]. We believe they are more readable.

Reviewer 2 Report
I think the paper is good and enough for publication. My decision is accept.
Please see the attachment

Author Response
Reviewer 2
Thank you very much for all the valuable comments that contributed to the improvement of the quality of the manuscript
The authors presented an article about “Promiscuous lipase-catalyzed Markovnikov addition of H-phosphites to vinyl esters for the synthesis of cytotoxic α-acyloxyoxy phosphonate derivatives”. I think the paper is well organized and appropriate for Materials journal but the paper will be ready for publication after minor revision.
Add more explanations for all tables and figures. They are inadequate.
There is an interesting approach and design exists, I just propose to emphasis the practical significance of the presented methodology in several points of article.
Response: Thank you for this remark. As suggested, we emphasized the importance of our approached toward synthesis of target compounds. The most important fact is that under developed protocol products without metal contamination are provided what fulfils very strict requirements of pharmacopeia regarding levels of metal impurities.
The success of the study needs to be identified as numerical.
Results and discussion and conclusion parts are inadequate according to citation and analyze in detail. There should be the importance of the study in detail, comparison results with other approaches in literature, the success of the prediction and computational results.
Improve the results and discussion and conclusion parts.
Please fix the typographical and eventual language problems in paper.
There is a reference problem. First, your reference list should contain more papers from Materials journal.
Response: all suggestions have been included in the manuscript and highlighted in green.

Reviewer 3 Report
The manuscript entitled “Promiscuous lipase-catalyzed Markovnikov addition of 2 H-phosphites to vinyl esters for the synthesis of cytotoxic 3 α-acyloxyoxy phosphonate derivatives” describes about the Markovnikov addition of H-phosphites to vinyl esters to produce α-acyloxy phosphonates with varying yields in the presence of Candida cylindracea (CcL) lipase catalyst. The production yield was tested against the changes in solvent, enzyme type, and temperature. The following points to be fixed before the work can be accepted for publication.
- The Introduction section is very much conservative. It will be good if the authors can include different sections consists of (a) Importance of α-hydroxy phosphonate derivatives as pharmacological agents, (b) their structure-activity relationships, (c) what makes them to use as potential antibacterial agents, (d) rationale behind the selection of α-acyloxy phosphonate derivative in the present study.
- Comparison of already available antibiotics and the need for the development of α-hydroxy phosphonate derivatives as potential pharmacological/antibacterial agents.
- The Introduction section should cover the importance of Candida cylindracea (CcL) lipase catalyst and Markonikov addition.
- Rationale behind the selection of Candida cylindracea (CcL) catalyst for this particular reaction is missing.
- Factors influencing the activity and reaction behavior of Candida cylindracea (CcL) catalyst.
- In Section 2.1, I don’t understand what is the importance of lines 65-86 here ? I didn’t realize that this section should include that particular information. The section is about “Microorganisms and Media”, but no information related to that is presented. Can the authors recheck this part ?
- The X- and Y-axes labels are missing in the images of Figures 3, 4, and 5.
- There is no need to provide a separate table just to show the statistically significant/insignificant data. Therefore, the Table 2 can be deleted and at the same time, this statistical information can be included in the corresponding images (next to the bars).
- Similar to Figures 3-5, Figure 7 also missing the axes labels in the image.
- The discussion need to be strengthened further. The comparison of results provided by samples 5-9 with that of similarly reported α-hydroxy phosphonate derivatives in the literature is required. Then only the readers know the importance of your synthesis and the formed products.
- Finally, I feel like the results provided are not just enough. The authors need to provide further analysis.

Author Response
Reviewers 3
Thank you very much for all the valuable comments that contributed to the improvement of the quality of the manuscript
The manuscript entitled “Promiscuous lipase-catalyzed Markovnikov addition of 2 H-phosphites to vinyl esters for the synthesis of cytotoxic 3 α-acyloxyoxy phosphonate derivatives” describes about the Markovnikov addition of H-phosphites to vinyl esters to produce α-acyloxy phosphonates with varying yields in the presence of Candida cylindracea (CcL) lipase catalyst. The production yield was tested against the changes in solvent, enzyme type, and temperature. The following points to be fixed before the work can be accepted for publication.
- The Introduction section is very much conservative. It will be good if the authors can include different sections consists of (a) Importance of α-hydroxy phosphonate derivatives as pharmacological agents, (b) their structure-activity relationships, (c) what makes them to use as potential antibacterial agents, (d) rationale behind the selection of α-acyloxy phosphonate derivative in the present study.
Response: The aim of our studies was to develop metal-free protocol toward hydroxy phosphonates with methyl group at alpha position and comparison with compounds possessing aryl group at the same group to revise the influence of this group on antibacterial activity against selected pathogenic E.coli strains. To our knowledge such correlation has not been done so far. As can be seen from our results methyl group increases activity of the tested compounds against selected patogens.
- Comparison of already available antibiotics and the need for the development of α-hydroxy phosphonate derivatives as potential pharmacological/antibacterial agents.
Response: Investigated compounds were compared with commonly used antibiotics ciprofloxacin (cipro), bleomycin (bleo), and cloxacillin (clox) please find Figure 7 in the Manuscript.
- The Introduction section should cover the importance of Candida cylindracea (CcL) lipase catalyst and Markonikov addition.
Response: Thank you for this remark. The Introduction was modified. In general application of enzymes as catalysts provides sustainable protocols which eliminates the need of usage of toxic metal catalysts. As a result of our studies we have developed enzymatic protocol which provide products without metal impurities. This is in line with the pharmacopoeial rules for metal contamination.
Rationale behind the selection of Candida cylindracea (CcL) catalyst for this particular reaction is missing.
Response: several enzymes including home-made liver acetone powder were tested. Due to the preliminary results CcL was selected as the most efficient catalyst. Tested enzymes are shown in Table 1 in the Manuscript.
- Factors influencing the activity and reaction behavior of Candida cylindracea (CcL) catalyst.
Response: Crucial factors like reaction temperature, type of solvent used as reaction medium, and amount of used enzyme were discussed. The influence of these factors on the course of the studied reactions was collected in Table 1.
- In Section 2.1, I don’t understand what is the importance of lines 65-86 here ? I didn’t realize that this section should include that particular information. The section is about “Microorganisms and Media”, but no information related to that is presented. Can the authors recheck this part ?
- The X- and Y-axes labels are missing in the images of Figures 3, 4, and 5.
- There is no need to provide a separate table just to show the statistically significant/insignificant data. Therefore, the Table 2 can be deleted and at the same time, this statistical information can be included in the corresponding images (next to the bars).
- Similar to Figures 3-5, Figure 7 also missing the axes labels in the image.
- The discussion need to be strengthened further. The comparison of results provided by samples 5-9 with that of similarly reported α-hydroxy phosphonate derivatives in the literature is required. Then only the readers know the importance of your synthesis and the formed products.
Response: We are grateful for this remark. The aim of this studies was to verified the influence of the methyl group located at the alpha position in the studied compounds which were compared with derivatives possessing aromatic group at he same position. Obtained results clearly showed that the compounds with smaller group at alpha position exhibit higher activity against pathogenic strains.
- Finally, I feel like the results provided are not just enough. The authors need to provide further analysis.
Response: all suggestions have been included in the manuscript and highlighted in green.

Round 2
Reviewer 3 Report
I see that the responses provided by the authors are ok to me. They are all not answered with high level of satisfaction, but ok to proceed depending upon the importance and scientific significance of the topic/research area that the authors selected. Therefore, I recommend this article for the publication.